# Stabilisation of a Plastic Soil with Alkali Activated Cements Developed from Industrial Wastes

Nuno Cristelo [1,*], Jhonathan Rivera [1], Tiago Miranda [2] and Ana Fernández-Jiménez [3]

[1] CQ-VR, Department of Engineering, University of Trás-os-Montes e Alto Douro, Quinta de Prados, 5000-801 Vila Real, Portugal; jhonathan@utad.pt
[2] ISISE, Institute of Science and Innovation for Bio-Sustainability (IB-S), Department of Civil Engineering, University of Minho, 4800-058 Guimarães, Portugal; tmiranda@civil.uminho.com
[3] Instituto Eduardo Torroja (IETcc), CSIC, C/Serrano Galvache 4, 28033 Madrid, Spain; anafj@ietcc.csic.es
* Correspondence: ncristel@utad.pt

**Abstract:** The development of alternative materials for the construction industry, based on different types of waste, is gaining significant importance in recent years. This is mostly due to the need to increase sustainability of this heavily polluting activity, thus mitigating the dependence on, for instance, Portland cement. The present paper is related to the development of an alkaline activated cement (AAC) exclusively fabricated from industrial by-products (both precursor and activator). Coal combustion fly ash, a common residue from thermoelectric powerplants, and glass waste, from the manufacture of ophthalmic lenses, were used as precursors. These precursors were activated with a recycled alkaline solution, resulting from the cleaning of aluminium extrusion dies, instead of the more common commercial reagents usually applied for this type of binder. Several pastes were studied, combining the precursor and alkaline solution in different proportions. When the most-performing cements were defined, they were used to stabilise a cohesive soil. The experimental procedure and subsequent analysis were designed based on a *Response Surface Methodology* model, considering the *Activator/Solids* and *Soil/Precursor* ratios as the most relevant variables of the stabilisation process. It was observed that, depending on the type of alkaline cement used, there was an optimum precursor and activator contents to optimise the mechanical properties of the stabilised soil. The reliability of this prediction was especially dependent on the type of precursors and, also, on their respective dissolution process right before the homogenization with the soil, under the working conditions available.

**Keywords:** alkali activated cements; sustainability; soil stabilisation; fly ash; glass waste





## 1. Introduction

The combination of more sustainable and efficient waste management, along with the interest of incorporating some industrial wastes in civil engineering applications, due to their specific properties, has seen a dramatic increase in recent years. In the matter of developing alternatives to Portland cement, this interest is mostly due to the exceptionally good results obtained when incorporating fly ash and blast furnace slag as supplementary cementitious materials (SCMs) or even as formal cementitious materials [1,2]. The development of this class of binders based on wastes, and the introduction of these new materials in engineering design, arises from the need to develop alternatives to the ever-present traditional calcium-based cements, as this dependence results in a very significant energy expenditure and generates high CO2 emissions. Indeed, the production chain of Portland cement is one of the industries generating the highest rates of greenhouse gases [3–5].

Alkali activated cements (AAC) are one of the alternative materials to traditional Portland cement that have attracted more attention in recent years [6–8]. The most relevant advantage of this type of material is its heavy incorporation of industrial wastes or by-products in its composition, which are used as base raw materials. Currently, the most

important and researched precursors/wastes are: different slags, coal fly ash, volcanic ash and glass waste. In theory, any material rich in aluminosilicates can be used to generate this new class of binder [9–15]. However, the current necessity of activators such as silicates and alkali metal hydroxides clearly constitute the weaker point regarding the synthesis of these materials, especially if the aim is to consider them as emerging substitutes for calcium-based binders. This significant drawback arises from the fact that the production of these reagents is also heavily polluting, while also raising the financial cost of these alternatives to similar levels of those found for traditional cements [16]. According to Habert et al. (2011) [17], the production of some of the alkaline activators can have a significant impact on the levels of toxicity in humans, water sources and soils. Therefore, in terms of research, it is necessary to work around this issue and try to find new alkaline activators with low levels of pollution associated. This undoubtedly would open the door to several new precursors to be used in the synthesis of new AAC, with more sustainable environmentally and economically performances compared with the current solutions.

Several research projects have been developed around this topic, mostly trying to obtain effective substitutes for traditional alkaline activators, usually by using residues from different sources. Studies carried out by Torres-Carrasco et al. (2014, 2015) [18,19] reported the solubility of various glass residues in alkaline solutions based on sodium, with the objective of synthesizing sodium silicate to use it as an alkaline activator in the production of binders. The temperature and particle size of the glass during dissolution were two important parameters controlling the quality of the final product. With a temperature of 80 °C and a particle size of 45 μm, the authors were able to dissolve up to 60% of the glass residue. The authors synthesized cements based on ash and glass residues, activated with NaOH solutions, which generated sodium silicate in situ and concluded that this type of residue can become an effective substitute of sodium silicate, to be used in AAC.

Fernández-Jiménez et al. (2017) [20] and Cristelo et al. (2019) [21] carried out sustainability studies regarding the use of various types of industrial wastes in the preparation of AAC, namely glass waste from the manufacture of optical lenses, sludge used in anodizing aluminium and coal fly ash. Additionally, a recycled alkaline solution, resulting from the cleaning of aluminium extrusion rods, was used for activation as a replacement of the common NaOH solution. The results showed that the cleaning solution could constitute an effective alkaline activator. The compressive strength after 28 days curing reached similar levels to those obtained with an 8M NaOH solution. When glass waste was also included, the cleaning solution performed better than the reference NaOH solution.

A different type of waste was used by Passuello et al. (2017) [22], when the authors carried out life cycle analyses of AAC based on calcined mine tailings and agro-industrial waste ash. Results showed that the cement synthesized from these wastes decreased $CO_2$ emissions by 70%, as compared with those released by Portland cement in a similar scenario. Furthermore, a reduction of more than 60% was observed in six of the nine environmental impact categories analysed in this study. Previous works have shown that some unconventional sources of silica and alkalis could integrate new processes for the manufacture of alkaline activators, with lower environmental impacts, while at the same time integrating the composition of new AAC, aiming the production and application of competitive and more sustainable alternatives to Portland cement-based binders.

In the present study, the effectiveness of newly developed alkali activated cements as a stabiliser of a plastic soil was assessed. The AAC were previously produced from glass waste (GW) recovered from the production of ophthalmic lenses, and type F coal fly ash (FA) from a thermoelectric powerplant. These precursors were activated with a recycled alkaline solution, originally used by the aluminium industry to clean extrusion rods. The quality assessment was focused on the mechanical performance of the cements, using uniaxial compressive strength tests. Two parameters were used to control the pastes, namely the FA/GW and the *Activator/Precursor* weight ratios, with the objective of defining the most effective combinations of the different wastes at disposal. To better characterise the progress of the alkaline activation reactions with curing time, seismic wave velocity

measurements were taken throughout the first 90 days of curing. The microstructure of the cement pastes was also characterised, using BSEM. After selecting the most performing cements, a response surface model was implemented with the objective of optimizing the soil-cement combinations, again using compressive strength as the main evaluation factor. It was considered important to determine if these models can be used with confidence, and if their precision and accuracy are satisfactory. The most effective soil-cement combination was then subjected to durability tests, considering different environments to evaluate the consequent volumetric and mass change. It is important to determine if alternative cements can indeed sustain their short-term behaviour throughout longer periods, based on their demanding solicitations and expected service life. Finally, the environmental performance of a chemically stabilised soil is vital for its potential field application, in this case even more so, considering the origin of the materials used to fabricate the stabilising cement. Therefore, a leaching test was performed on a selected combination, with the objective of detecting any excessive leaching of heavy metals present in the original wastes.

## 2. Materials and Methods

### 2.1. Materials

The materials used for the synthesis of the cements were residues from various industrial sources. A type F fly ash (FA) from the Portuguese thermal power plant of 'Pego' and glass waste (GW) from a Portuguese company dedicated to the production of different types of optical lenses were used as precursors. Table 1 presents the chemical composition of these two precursors. The fly ash composition is very common in this type of coal combustion residue, i.e., a material rich in aluminosilicates. According to its chemical composition, and based on the criteria of the ASTM C618 standard [23], it can be classified as a type F fly ash. The chemical composition of the glass waste reveals a material with a significant lead content, which is expected from the glass generally used for the optical elements. The chemical composition of the precursors was obtained by x-ray fluorescence with a PHILLIPS PW-1004 spectrometer.

**Table 1.** Elemental composition of the precursors.

| Element | WG | FA |
|---|---|---|
|  | (wt%) | (wt%) |
| $Na_2O$ | 8.750 | 1.180 |
| $SiO_2$ | 58.37 | 54.77 |
| $Al_2O_3$ | 3.940 | 21.57 |
| MgO | 0.492 | 1.870 |
| $K_2O$ | 4.659 | 2.559 |
| CaO | 6.128 | 1.680 |
| $TiO_2$ | 2.730 | 1.010 |
| $Fe_2O_3$ | 0.148 | 7.325 |
| $SO_3$ | - | 0.66 |
| ZnO | 2.572 | 0.026 |
| $ZrO_2$ | 2.155 | 0.045 |
| BaO | 2.300 | 0.121 |
| PbO | 5.072 | 0.004 |
| $Cr_2O_3$ | - | 0.06 |
| L.O.I. | 1.849 | 6.470 |

The recycled cleaning alkaline solution (CS) used as an alkaline activator was retrieved in liquid form, with a pH > 13 and a density of 1.3 g/cm3. It was subjected to a previous homogenization process, as solids were sedimented in the bottom of the container upon arriving to the laboratory. Table 2 presents the elemental composition of the recycled solution, obtained by Inductively Coupled Plasma Atomic Emission Spectrometer (ICP-AES), with a plasma power of 1.4 kW, plasma gas flow 15.00 L/min, nebulizer gas flow 0.89 L/min with a reading time of 5 s.

**Table 2.** Elemental composition of the recycled cleaning solution.

| $Al_2O_3$ | $Na_2O$ | $SO_3$ | $H_2O$ | pH | [a][OH]$^-$ | Density (g/cm$^3$) |
|---|---|---|---|---|---|---|
| 7.14 | 12.13 | 1.18 | 79.5 | >13 | 5.3 | 1.30 |

[a] Acid-base reaction with 5 N HCL (Panrea S.A.).

The mineralogical composition of the precursors was characterized by X-ray diffraction (Figure 1). A PANalytical X'Pert Pro diffractometer was used, equipped with an X'Celerator detector and a secondary monochromator, covering an interval between 5 and 60°(2θ), with a nominal step size of 0.017° and 100 s/step. The diffraction spectra shows quartz and mullite as the most representative minerals of the fly ash, as well as an halo between 17–33°(2θ). This halo reveals the presence of amorphous content in fly ash, which is characteristic of semi-crystalline materials. The glass diffractogram reveals a purely amorphous material, given the absence of the peaks characteristic of crystalline phases.

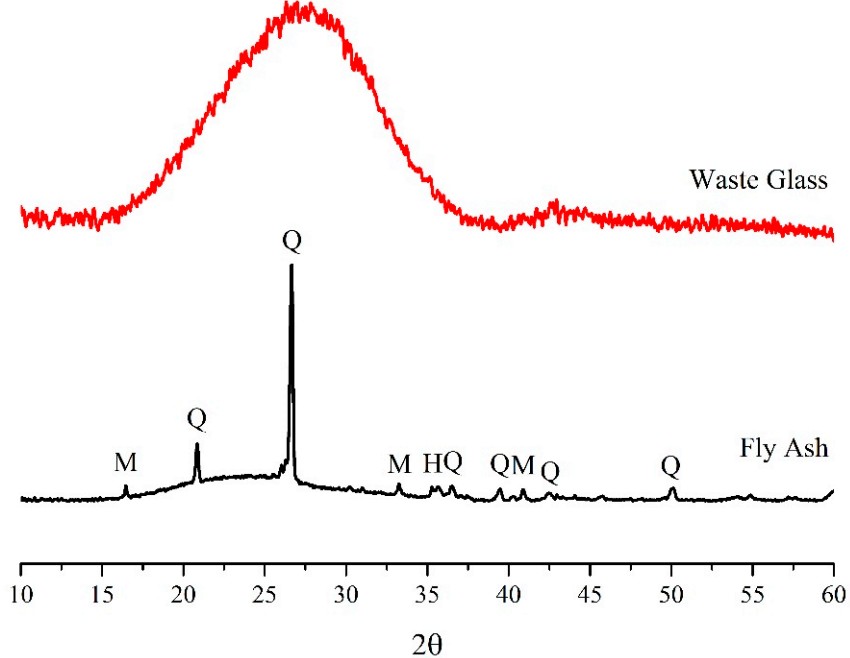

**Figure 1.** Predominant minerals in the precursors: H = hematite ($Fe_2O_3$); Mu = mullite ($3Al_2O_3+2SiO_2$); Q = quartz ($SiO_2$).

The soil was collected in the north of Portugal, near the city of Chaves, home of a well-known industry cluster dedicated to the fabrication of clay bricks and roof tiles, taking advantage of the local clayey soil. The particle size distribution of the soil, fly ash and glass waste is presented in Figure 2. It was obtained after drying the materials, using laser diffraction, on a Sympatec Helos BF particle size analyser, with a measurement range between 0.9–117 mm. The weight percentage of the particles smaller than 45 μm was 81.3 wt% (soil), 75.6 wt% (fly ash) and 99.2 wt% (glass). The particle size of the precursors is acceptable, especially considering that they have not been subjected to any additional milling process. In general, the lower the particle sizes of the precursor, the higher its reactivity, which benefits the dissolution process under the alkaline environment created by the activator.

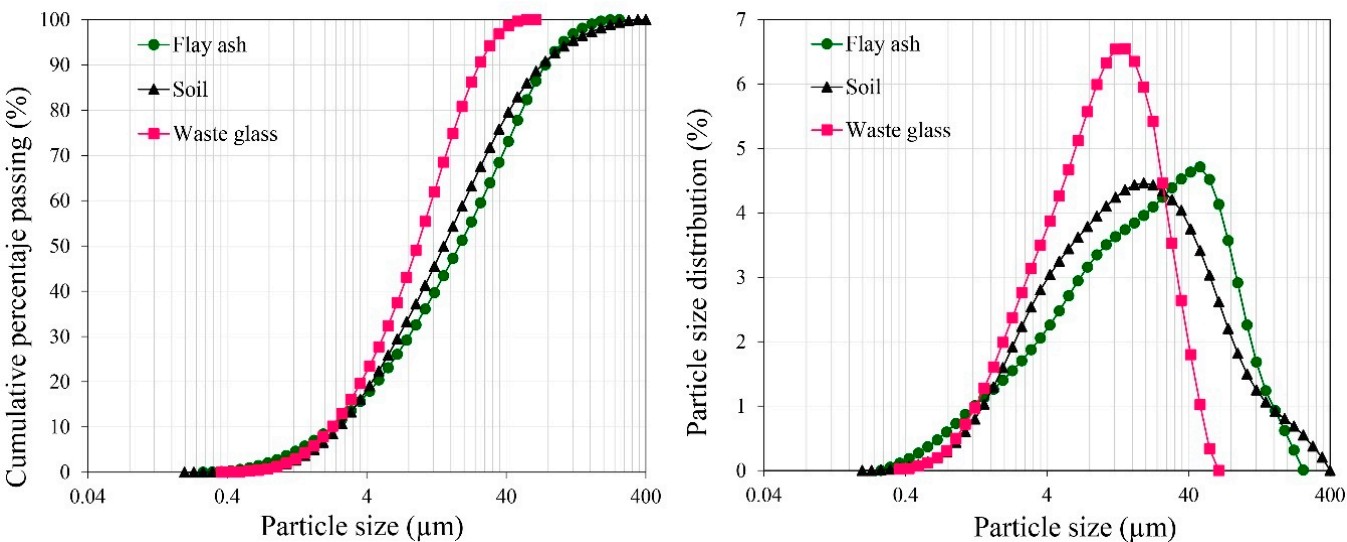

**Figure 2.** Particle size distribution of the original soil, fly ash and glass waste.

### 2.2. Preparation and Testing of the Cements

The initial formulations of the cements combined the two precursors–fly ash and glass waste–in percentage ratios of 100/00, 75/25, 50/50, 25/75 and 00/100, respectively. The weight ratio between the activator and the solids in the mixture varied between 0.4 and 0.85, aiming to optimise the workability of each mixture. The identification and composition of the cements is presented in Table 3. Preparation of the pastes began with the homogenization of the dry precursors' mixture, followed by the addition of the alkaline solution and subsequent further mixing, for 3 minutes. The fresh pastes were then poured into 40 × 40 × 40 mm moulds.

**Table 3.** Composition of the alkali activated cements.

| AAC ID | Precursor (wt%) | | Activator/Precursor |
|---|---|---|---|
| | **FA** | **WG** | **(wt. ratio)** [1] |
| **100FA** | 100 | - | **0.40** |
| | | | 0.47 |
| | | | 0.55 |
| 75FA25WG | 75 | 25 | 0.45 |
| | | | 0.52 |
| | | | 0.65 |
| **50FA50WG** | 50 | 50 | **0.50** |
| | | | 0.57 |
| | | | 0.75 |
| 25FA50WG | 25 | 75 | 0.55 |
| | | | 0.62 |
| | | | 0.80 |
| **100WG** | - | 100 | **0.60** |
| | | | 0.67 |
| | | | 0.85 |

[1] The *Activator/Precursor* ratios obtained at this stage of the study were not transferred to the next stage (regarding the application of the AACs to the soil stabilisation), as the total liquid phase during that next stage was defined based also on the soil's humidity requirements. [2] Pastes in bold were selected for further testing, after analysis of the uniaxial compressive strength.

During the first 72 h, the pastes were cured still inside the moulds, under temperature/humidity conditions of 50 °C and 25%, respectively, in a climatic test chamber. These curing conditions were chosen with the intention to simulate, as much as possible, the field curing conditions. After this initial curing period, the specimens were removed from the mould and left to cure under the same temperature and humidity conditions, until the scheduled tests. The compressive strength tests were carried out after a total of seven days curing, in a servo-hydraulic bench top load frame equipped with a 25 kN capacity load cell.

Samples were collected from specific pastes for microstructural characterisation, using scanning electron microscopy coupled with a backscattered electron detector. These analyses were performed on a Hitachi S-4800 scanning electron microscope (20 kV), in low vacuum mode (1.3 mbar), avoiding the deposition of a conductive layer. The device was fitted with a solid-state BSE detector and a X-ray energy dispersive analyser (EDX)–Oxford LINK-ISIS–using a ZAF correction model for quantitative chemical analysis.

After testing the different formulations, 3 specific precursor combinations were chosen for further analysis and for subsequent application to the cohesive soil stabilisation process. The two extreme formulations were selected (100FA and 100WG), together with the formulation representing the equal proportion of the two precursors, i.e., the 50FA50WG. The most effective *Activator/Precursor* ratio for each of the three selected pastes was chosen, for further analysis of the pastes. However, for the subsequent study of the soil-cement combinations, these *Activator/Precursor* ratios were disregarded and substituted by the *Activator/Solids* ratio (with '*Solids*' = *activator + soil*). This is because the addition of dry soil demanded an increase in the liquid phase of the mixture, which was obtained by increasing the Activator content and, thus, modified the initial *Activator/Precursor* ratios of the cements.

A dynamic compaction test (Modified Proctor Compaction test), based on the contents of the standard [24], was developed simultaneously for each of the three selected pastes, combined with soil. Essential information was thus obtained regarding the optimum moisture content for each of the soil-binder mixtures, which was used to design the experimental program. Such experimental program was defined using the Response Surface Methodology (RSM), which allowed the optimisation of the total number of uniaxial compressive strength tests. The RSM model was designed with two variables, namely the Soil/Precursor and the Activator/Solids weight ratios (in this case, the 'solids' was considered to be the total sum of the precursor and soil weights).

### 2.3. Seismic Wave Velocity

Non-destructive ultrasonic pulse velocities were measured on 15 cm cubic specimens of the cements 100FA (*Activator/Precursor* = 0.40), 50FA50WG (0.50) and 100WG (0.60), up to a period of 80 days, using frequencies specifically targeting the vibration of the piezoelectric transducers in a way as to produce compression-type waves (P-waves).

A set of two piezoceramic bender-extender transducers were already in position inside the mould when the fresh paste was poured and vibrated. The negatives left by the bender elements, when removed from the specimens after the measurements were finished, can be seen in Figure 3. One of the bender elements was used to transmit the P-waves through the soil, while the other was prepared to receive those same waves. Commercially available equipment was used to generate, amplify and acquire the signals, namely a pulse waveform generator, an output amplifier and an oscilloscope.

The signals were transmitted in the horizontal direction with varying excitation voltages and frequencies, with the constant intention of finding the most adequate energy/frequency combination to the evolutionary state of the specimen, at the time of each measurement (i.e., as the curing developed, the physical status of the material evolved, thus responding differently to any given frequency that was very effective in the last measurement). The exact travel length and remaining two measures of the cube, as well as the weight of each specimen, were measured before each reading (to account for volumetric and water content variations). The acoustic coupling between the transducers and the

surrounding specimen was improved with a layer of ultrasound conductive gel, applied at the time of the moulding of each specimen.

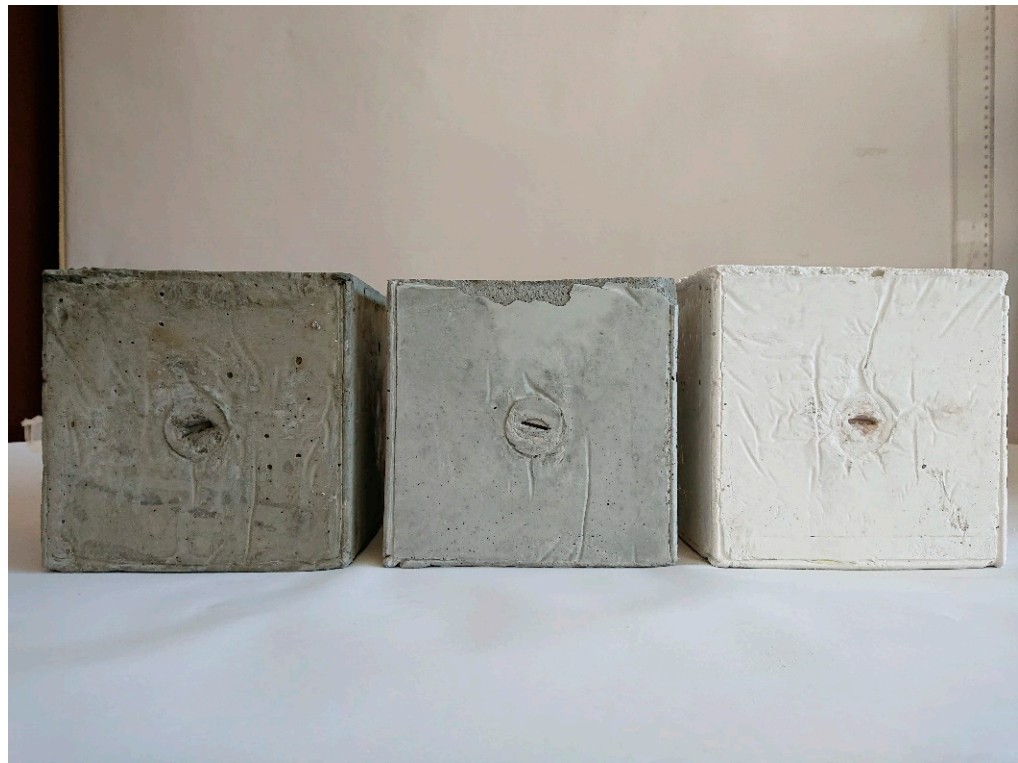

**Figure 3.** Cubic specimens of the binders 100FA (**left**), 50FA50WG (**centre**) and 100WG (**right**), showing the insertion points of the bender elements which used to measure the seismic waves.

Using software purposely developed by the company providing the oscilloscope, both the emitted and received waves could be seen real-time, as well as the flight time of the wave (if desired, a set of cursors could be positioned on each wave, and the oscilloscope yielded the time difference between the two positions). The oscilloscope was always connected to a PC, which allowed the collection and storage of the selected data.

*2.4. Durability Tests*

Wetting/drying durability tests were performed on the soil stabilised with the 50FA50WG, to assess the medium-term quality of this solution. The ASTM D559 [25] procedure, designated by "Test Method B (soil < 19 mm)", was followed. It consists on the measurement of the mass, volume and moisture variations of the stabilised soil throughout 12 cycles. Two specimens of each mixture were submitted to these cycles; one was brushed after each cycle with a hand-made force of approximately 13.3 N, while the volume change and moisture content was measured on the other. After curing for 28 days, the specimens were submerged in water, at room temperature, for 5 h, followed by a period of 42 h in an oven at 71 °C. These 47 h constituted a full cycle. Three different types of test were performed, using different environments for the wetting phase of the test: alkaline, acid and neutral. The alkaline environment was obtained by diluting sodium carbonate in water until a pH of 10 was reached. Sulfuric acid diluted in water was used to obtain an acid environment with a pH of 4. Finally, tap water was used for the neutral environment. The carbonate or acid were added to the water where the specimen was to be submerged until the desired pH was reached. The solution was renewed for each new cycle, in order to keep the pH of the medium constant, during the test.

*2.5. Leaching Tests*

To evaluate the contamination risk associated with the use of wastes in the soil stabiliser, the same combination of soil and 50FA50WG used for the durability tests was also submitted to a leaching analysis, following the compliance test proposed by the EN 12457-4 (2002) [26]. This is a one stage batch test at a liquid/solid of 10 L/kg, using particles smaller than 10 mm. The specimen used for the leaching test was fabricated in a similar way as those used for the compressive strength tests. After 28 days curing, this specimen was crushed and sieved through a 10 mm mesh. A particle sample of 90 g was then mixed with 900 kg of water, at 10 rrpm., for 24 h, after which the eluate was recovered. The chemical analysis of the eluate was compared with the limit values proposed by the Council Decision 2003/33/CE (2003) [27], for inert and non-hazardous wastes. The analysis focused on the most dangerous elements with a significant presence in the original wastes, namely lead, zinc and sulphates.

## 3. Results and Discussion

The study described was divided in two main stages. The first stage focused on the design of the alkaline cements, using the uniaxial compressive strength as the main reference, supported by backscattering electron semi-quantitative characterisation, X-ray diffraction and seismic wave velocity. The second stage focused on the soil-cement performance, using the three most performing alkaline cements from the previous stage. This second stage was designed using a Response Surface Methodology model, to determine the maximum compressive strength, and included durability and environmental assessments of the optimum soil-cement combination.

*3.1. First Stage: Design and Characterisation of the Cements*

3.1.1. Compressive Strength of the Alkaline Cements

The compressive strength results of the alkali activated cements, after 7 days curing, are shown in Figure 4. The UCS obtained with the lowest *Activator/Precursor* ratios of the 100FA, 75FA25WG and 50FA50WG combinations was similar. When the fly ash content dropped below to 25 wt%, the UCS decreased significantly. However, even for the cements prepared only with waste glass (100WG), the UCS was approximately 9 MPa, which is acceptable for certain applications. In general, the best results were achieved with the lowest *Activator/Precursor* ratio of each combination, but this lowest ratio was different for each cement, increasing as the FA content in the precursor decreased. For the subsequent soil stabilisation tests, the extreme compositions, i.e., 100FA and 100WG, given the fact that the former reached the highest UCS and the latter used only waste glass (a less versatile waste in terms of Civil Engineering applications), also reached an acceptable UCS. A third cement was selected as a compromise solution—50FA50WG. The lowest *Activator/Precursor* ratio of each of the three selected pastes was chosen for further analysis of the pastes, i.e., 0.40 (100FA), 0.50 (50FA50WG) and 0.60 (100WG).

3.1.2. BSEM of the Alkaline Cements

The ternary diagrams presented in Figure 5 constitute a semi-quantitative analysis of the reaction products on the gel phase of each of the three selected mixtures, obtained with BSEM. The reaction products of the 100FA mixture are characteristic of sodium aluminosilicates of the type $SiO_2$-$Al_2O_3$-$Na_2O$, with little or no CaO content. With the increase in waste glass content, a slight increase in CaO can be observed in the composition of the reaction products, a natural consequence of the original CaO content in this precursor, higher than that found in the fly ash Table 1.

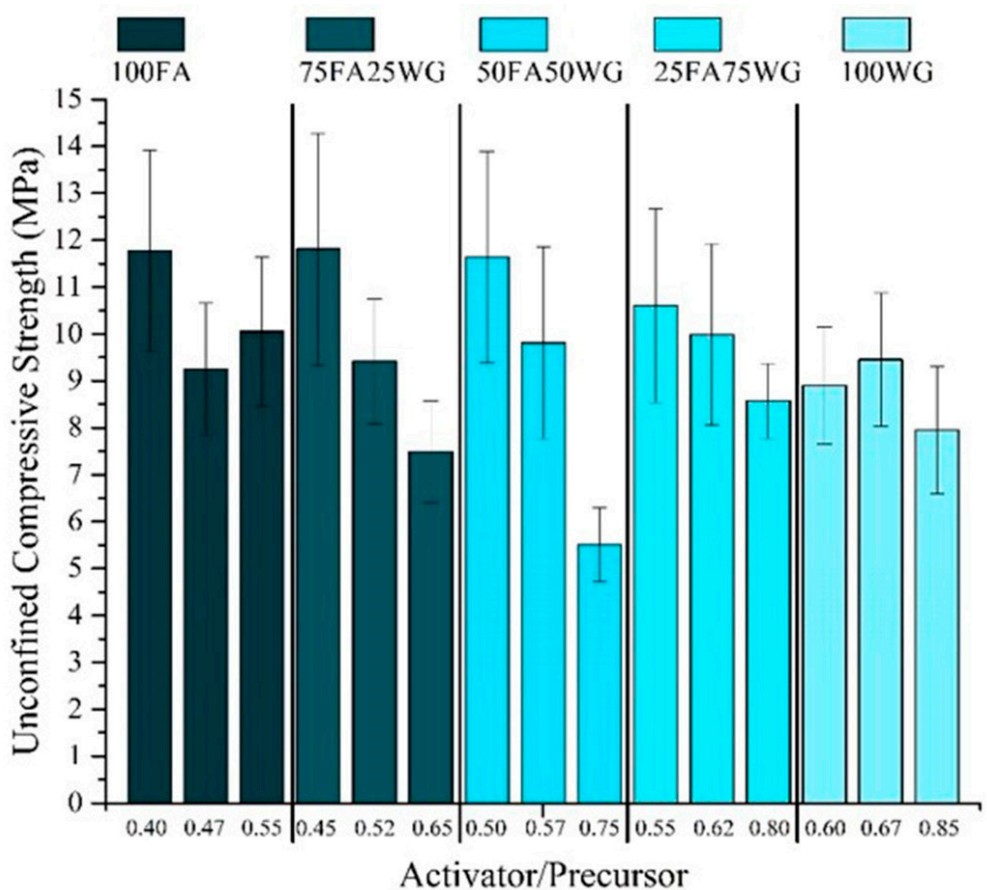

**Figure 4.** Unconfined compressive strength of the pastes after 7 days curing.

### 3.1.3. XRD of the Alkaline Cements

The X-ray diffraction data of the cements after 28 days curing is presented in Figure 6. Some of the original phases of the FA, such as Quartz and Mullite, were also identified in the 100FA cement. After the activation with the recycled solution, Sodalite-type zeolites ($Na_4Al_3Si_3O_{12}Cl$), a type of zeolite commonly developed in alkaline cements based on coal fly ash, was identified in all pastes [28,29]. Thermonatrite ($Na_2CO_3+H_2O$) was also identified in the 100WG cement. Being a type of sodium carbonate, it is possible that it was generated due to an excess of activator in the formulation. Another possible explanation is the low solubility of the waste glass under these conditions, allowing the sodium compounds in the recycled solution to remain free and, consequently, available to react with the atmospheric $CO_2$, generating this type of carbonate [30].

### 3.1.4. Seismic Wave Measurement of the Alkaline Cements

Figure 7 presents some examples of the received waves, for each of the three pastes, measured after three different curing periods. The signals presented were obtained with the most effective frequency and peak-to-peak amplitude for that particular paste and curing time.

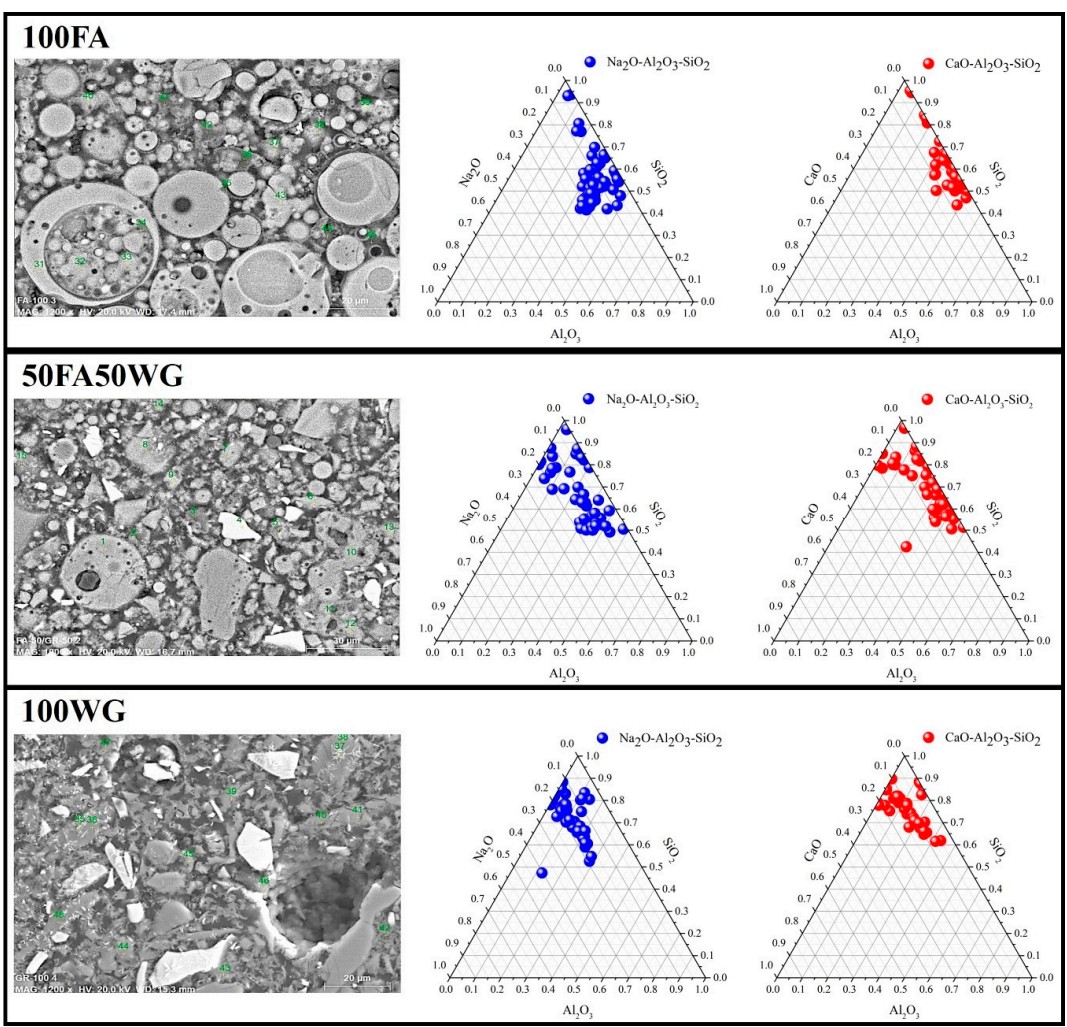

**Figure 5.** BSEM micrographs and corresponding EDX points of the pastes 100/00, 50/50 and 00/100 after 7 days curing.

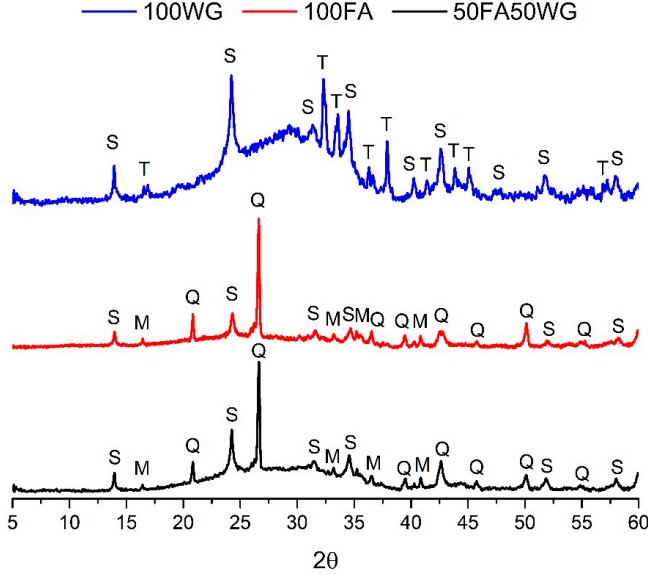

**Figure 6.** Predominant minerals in the selected pastes, after 28 days curing: M = mullite ($3Al_2O_3+2SiO_2$); Q = quartz ($SiO_2$); S = sodalite ($Na_4Al_3Si_3O_{12}Cl$); T = thermonatrite ($Na_2CO_3+H_2O$).

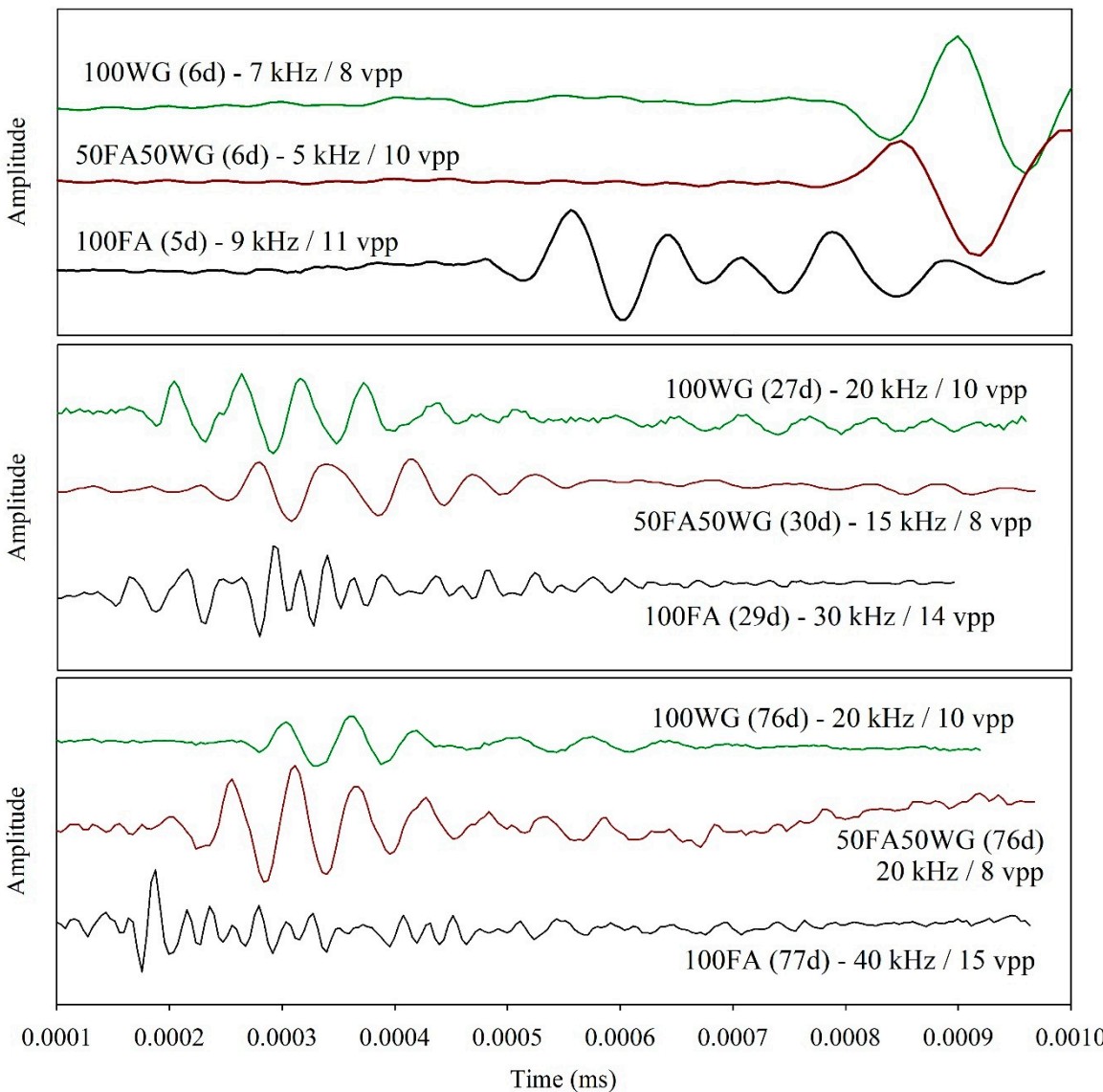

**Figure 7.** Examples of P-wave signals measured in pastes 100FA, 50FA50WG and 100WG after different curing periods.

Some trends were clear throughout the curing range. The fresh/slurry initial state of the material produced very low initial frequencies, as high frequencies produced flat response signals during this initial period. The progressive hardening of the pastes with curing time resulted, as expected, in an increase in the optimum frequency of each paste [31].

Such behaviour is more evident as the fly ash content increases, which might be explained by the well-known slow reaction rate obtained when coal fly ash type F is used as a precursor, as was the case. The optimum frequency for the 100FA paste increased from 9 kHz to 30 kHz, between the 5th and 29th curing days, and increased again, to 40 kHz, after the 77th day. The optimum measuring frequency for the 50FA50WG paste also increased with curing time, but less significantly–from 5 kHz (6th day) to 15 kHz (30th day), and again to 20 kHz, after day number 76. Finally, the optimum measuring frequency of the 100WG paste increased only during the first curing period, from 7 kHz (6th day) to 20 kHz (27th day), which was also the optimum frequency after 76 days.

The response amplitudes, on the other hand, didn't present a similar behaviour, as the values registered with the higher frequencies (for the longer curing periods) were very similar to those obtained during the first curing stages, i.e., between 8 and 10 vpp for the pastes with waste glass, and between 11 and 15 ppm for the 100FA paste. This is in

accordance with the findings obtained by Trtnik and Gams [31], when using P-waves to monitor the setting of Portland cement pastes.

The trend, in terms of optimum frequency, is in accordance with the corresponding P-wave velocities shown in Figure 8, i.e., pastes 100FA and 50FA50WG showed a progressive velocity increase right until the last measurement (77 and 78 days, respectively), while the P-wave of the 100WG paste stabilised (and even decreased) after 27 days curing. This might be an indication of the higher porosity of the 100WG pastes, as concluded by Zou and Meegoda [32], when trying to assess the porosity of dry and saturated Portland cement pastes based on P-wave velocity. It might also be related with the cracking evolution, which was, apparently, more intense in this last paste. The increase, with curing time, in the P-wave velocity (which is a direct measure of the elastic stiffness of the material) of the 100FA and 50FA50WG pastes is well in accordance with the usually observed strength increase of fly ash-based alkaline cements [33].

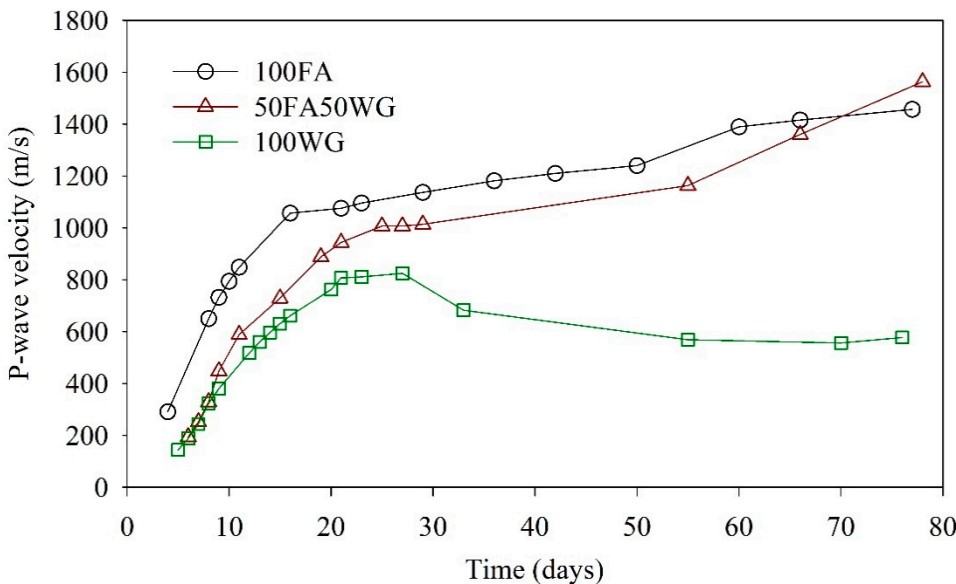

**Figure 8.** Evolution, with curing time, of the P-wave velocities in the three pastes.

*3.2. Second Stage: Soil-Cement Performance*

3.2.1. Compaction Parameters of the Soil-Cement

In order to study the soil-cement combinations using the 100FA, 50FA50WG and 100WG precursors, it was necessary to first establish the most effective compaction parameters. This was necessary because chemically stabilised soil is a friable material, which usually requires compaction. A series of Proctor tests, using standard compaction energy, were carried out, following the protocols of the standard BS 1377-4 [34]. Three Proctor tests were performed, for each of the three types of AAC chosen to stabilise the soil. The results, presented in Table 4, were used to define the expected *maximum dry density* of the specimens and, also, the maximum limit of the *optimal water content* compaction ranges, which were later introduced in the design of the soil-stabiliser experimental program (Response Surface Model, Section 3.2.2). The increase in *optimal water content* with the decreasing FA and increasing WG contents was probably related with the particle size of each precursor. The latter is finer than the former, thus requiring more liquid to lubricate the particles and facilitate the compaction process.

**Table 4.** Maximum Dry Density and Optimum Water Content for each Soil + AAC combination, obtained from the Proctor tests.

| Variable | Soil + 100FA | Soil + 50FA50WG | Soil + 100WG |
|---|---|---|---|
| Maximum dry density | 1.97 g/cm$^3$ | 1.95 g/cm$^3$ | 1.97 g/cm$^3$ |
| Optimum Water Content | 9.6% | 10.9% | 11.9% |

3.2.2. Response Surface Modelling of the Soil-Cement Combinations

The experimental design of the soil stabilisation process was modelled using the Response Surface Methodology (MSR), aiming to optimize specific variables, with a significant influence on the effectiveness of the stabilisers. The independent variables considered were the *Activator/Solid* (A/S) and *Soil/Precursor* (S/P) weight ratios, and the response variable was the compressive strength, after 28 and 180 days curing. Table 5 presents the values considered for each level, for each independent variable, for each of the Soil + AAC combination. The values for the S/P variable were defined from the minimum and maximum precursor content used (values defined based on experience), while the range for the A/S variable was defined based on the Proctor curve of each Soil + AAC combination. The maximum value of this A/S interval was very close to the *optimal moisture content* (note: this value was included in the mentioned range), while the minimum value was defined along the dry side of the respective Proctor curve, with the aim of limiting the water content to the minimum possible.

**Table 5.** Levels for each independent variable, for each Soil + AAC combination.

| Variable | Level | Soil + 100FA | Soil + 50FA50WG | Soil + 100WG |
|---|---|---|---|---|
| Soil / Precursor | Upper | 4.00 | 4.00 | 4.00 |
| (S/P) | Lower | 2.33 | 2.32 | 2.32 |
| | Central Point | 3.16 | 3.16 | 3.16 |
| Activator/Solids (*) | Upper | 0.10 | 0.11 | 0.12 |
| (A/S) | Lower | 0.05 | 0.06 | 0.08 |
| | Central Point | 0.07 | 0.09 | 0.10 |

(*) In this case, the term 'Solids' refers to the total sum of the precursor and soil weight.

A Central Compound Design (CCD) was implemented, with a total of 13 runs for each Soil + AAC combination. Each of these runs comprised 4 points on the hypercube, 5 central points and 4 axial points. The CCD is a rotary design, and its axial space α is considered as $(F)^{1/4}$, for k = 2 factors, $\alpha = (2^2)^{1/4} = 1.414$. The model was implemented and developed using the statistical analysis software *Minitab* (v17).

Figure 9 shows the graded surfaces generated by the values of the response variable (UCS after 28 and 90 days curing) for each of the three Soil + AAC combinations. These graphics allow an easy reading of the influence of the independent variables, and of the curing time, on the compressive strength of the stabilised soil. When using the 100FA cement, and after 28 days, the highest UCS was observed for the minimum and maximum value of the A/S and S/P variables, respectively, meaning that the compressive strength increases with the activator and precursor contents in the mixture [9,35]. On the other hand, with the increase in curing time (after 180 days), it seems that lower activator and precursor contents are required to reach UCS values similar to those obtained after only 28 days, suggesting that, with more time, the reacted FA increases, resulting in significantly higher UCS. This is a characteristic response of fly ash-based alkaline cements [16,36,37].

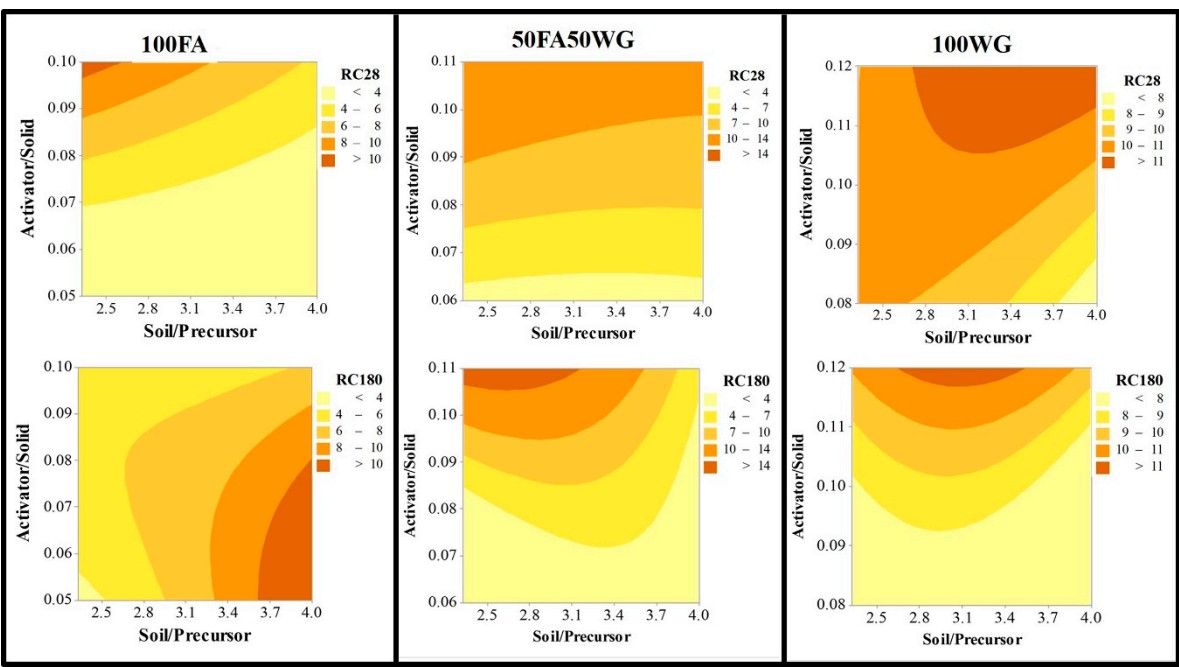

**Figure 9.** Response surfaces for compressive strength of the soil + cements, after 28 and 90 days curing.

The plots also show that, when stabilizing the soil with the 100WG cement, and after 28 days, the UCS depends, to a great extent, on the activator content. Indeed, when higher activator contents are used, the compressive strength is more or less constant, regardless of the precursor content. With the increase in curing time (i.e., after 180 days), a similar trend was observed, with the highest UCS values obtained with high activator contents and intermediate precursor contents.

Lastly, the application of the 50FA50WG cement revealed an increase in the unconfined compressive strength, compared with the results obtained with the other two AAC. The overall shape of the 28-day surface suggests that the activator content is decisive, with higher A/C values producing the highest UCS values, regardless of the precursor content. However, after 180 days, the strength of the stabilized soil seems more dependent on both the activator and precursor contents, which can be interpreted as a necessity to increase the quantity of cement so that it binds and connects most of the soil particles.

The individual optimization of each variable, for each soil-cement combination, was then pursued, aiming to experimentally validate the statistical results. Therefore, additional specimens were purposely prepared using the compositions which the model considered to be the most effective, according with Figure 9 and, more clearly, with Figure 10:

- 100FA: S/P = 2.33 and A/S = 0.100;
- 50FA50WG: S/P = 2.33 and A/S = 0.115;
- 100WG: S/P = 3.58 and A/S = 0.120.

The UCS of these additional specimens was tested after 28 days curing, and the results compared with the numerical values given by the model (Figure 10). The 50FA50WG and 100WG experimental mixtures registered significantly lower UCS values than the numerical values (64% and 58%, respectively). On the other hand, the 100FA experimental mixture registered a slightly higher UCS than the value predicted by the model (approximately 105%).

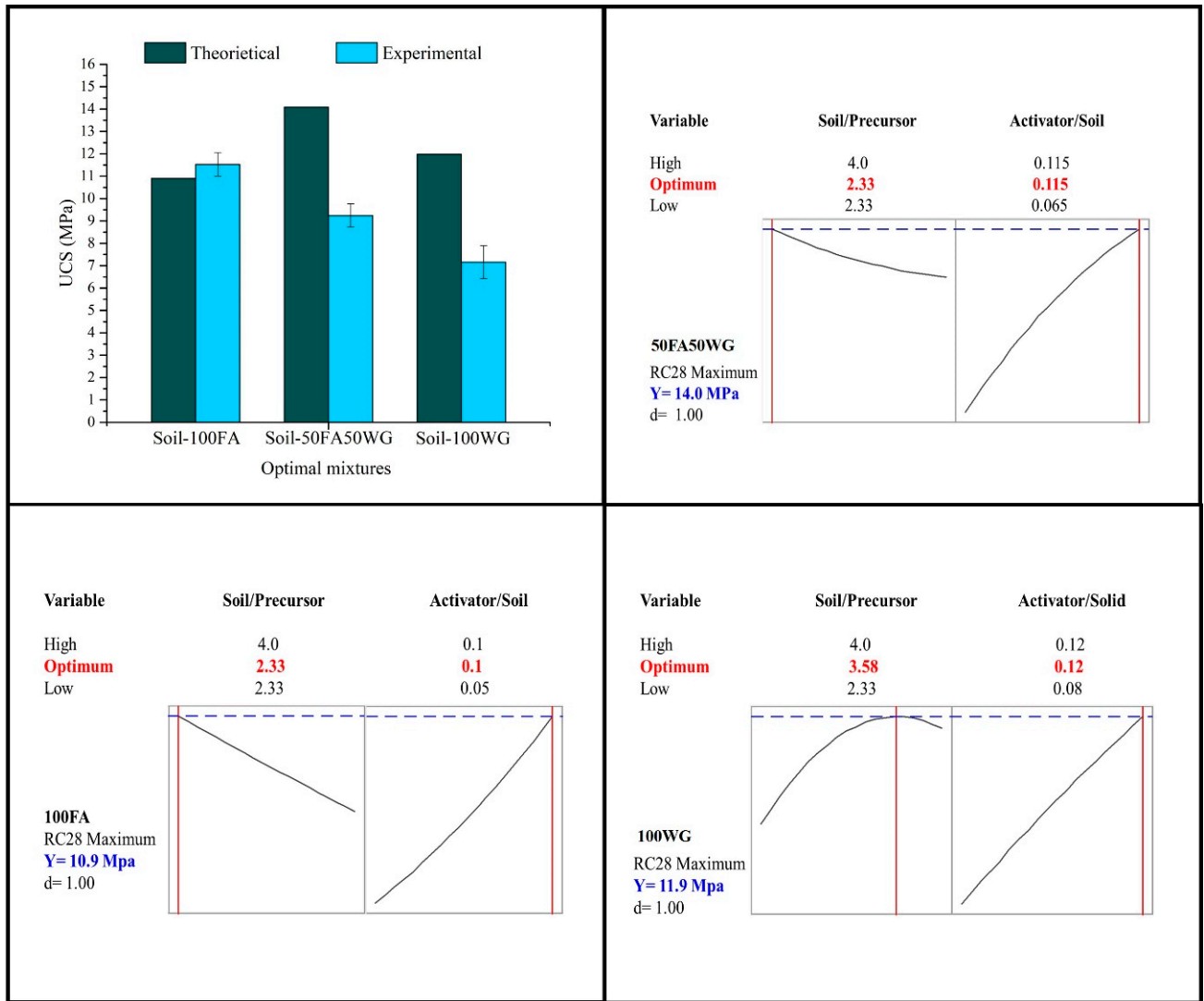

**Figure 10.** Prediction compressive strength profiles of the ideal formulation for each of the three soil + binder combinations tested, after 28 days curing—comparison with the experimental results.

The Coefficient of Determination (adjusted $R^2$) of the mixtures from the 50FA50WG and 100WG combinations was 80.99% and 67.94%, respectively (Table 6). This value defines the statistical significance of each of the models used for this experiment [38]. On the other hand, the predicted $R^2$ is also a measure of the predictive capacity of the regression models. As can be seen in Table 6, the predictive capacity of the 100WG model was low (57.24%), contrasting with that of the 100FA (83.16%) and 50FA50WG (73.6%) models. It is interesting to note that the 50FA50WG and 100WG cements have in common the use of glass waste, which has a very low solubility at low temperatures [18,39]. Therefore, it is possible that the composition of the precursors can hinder the effectiveness and precision of the prediction models.

**Table 6.** Response Surface Methodology model coefficients.

| Coefficient | Soil + 100FA | Soil + 100WG | Soil + 50FA50WG |
|---|---|---|---|
| $R^2$ | 88.38% | 84.79% | 73.52% |
| $R^2$ Exp. | 86.57% | 80.99% | 69.94% |
| $R^2$ Pred. | 83.16% | 73.60% | 57.24% |

Based on the results presented, the precursor 50FA50WG was selected for further characterisation of the soil-cement material. Although the experimental optimised UCS of the soil-cement fabricated with the 50FA50WG was lower than that obtained with the 100FA, the UCS of both cements was very similar (Figure 4), and the 50FA50WG allows the incorporation of a different type of waste (glass), with less known Civil Engineering applications than the fly ash. The optimised soil-cement composition presented in Figure 10 was also maintained for the subsequent studies regarding durability and environmental performance.

### 3.2.3. Durability of the Selected Soil-Cement Combination

Durability tests (wetting/drying cycles) were performed on the soil stabilised with the paste 50FA50WG, using the optimised composition of S/P = 2.33 and A/S = 0.115. These values represent weight contents of 70% soil and 30% precursor (the latter corresponds to 15% FA and 15% WG); and 90% solids (soil + precursor) and 10% activator, respectively.

Volumetric and mass changes were registered throughout 12 cycles. Three sets of 12 cycles were performed, using three different solutions for the wetting phases: neutral (pH = 7), acid (pH = 4) and alkaline (pH = 10). The volumetric changes and visual observation of the evolution of the specimens (Figure 11) denote an apparently similar influence of the different environments on their physical integrity, with the exception of cycle # 3 (alkaline medium) and cycle # 9 (neutral medium), when both presented a considerable material loss, which generated a very different reading relatively to the average volumetric change values.

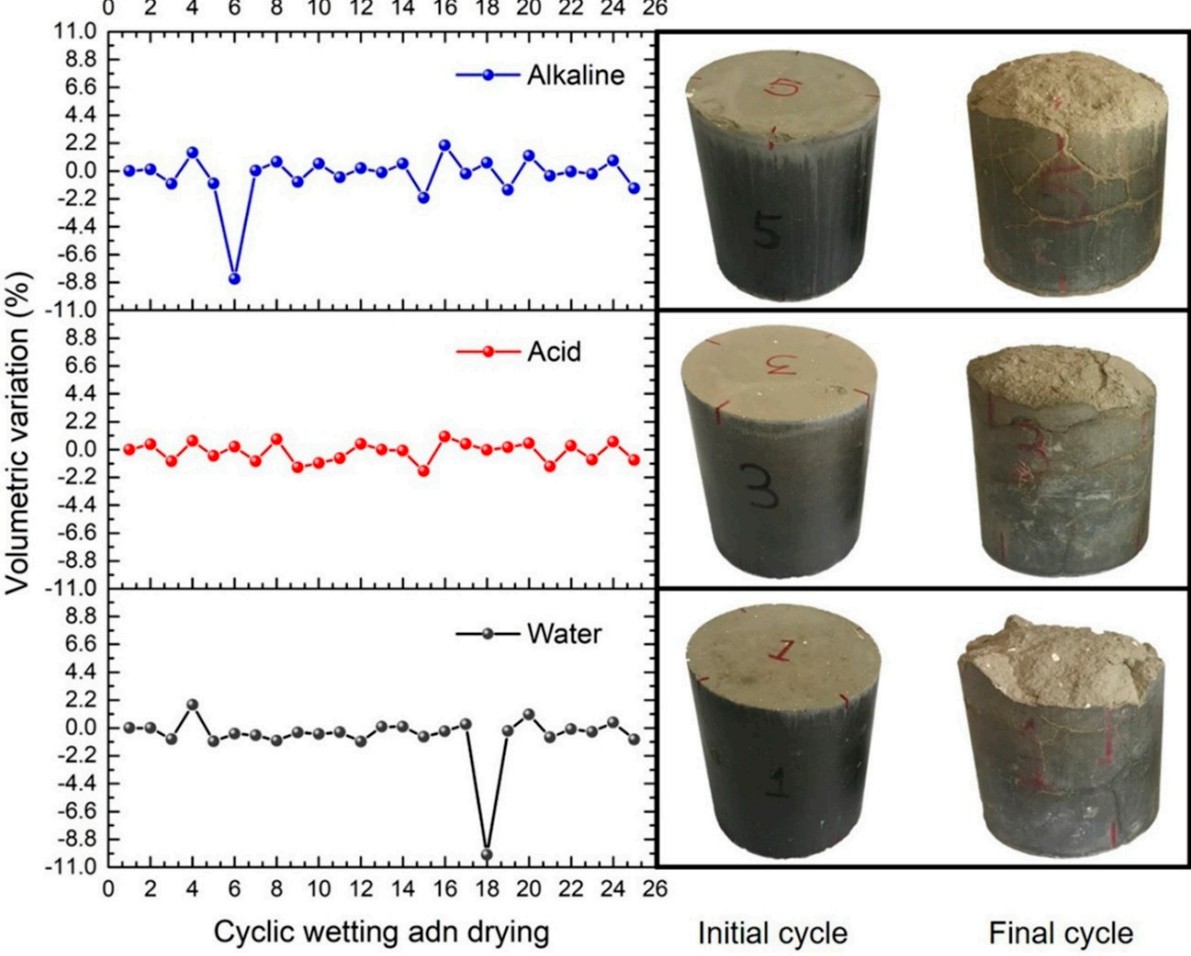

**Figure 11.** Volumetric changes during the wetting/drying cycles under alkaline, acidic and neutral medium.

The average volumetric expansion values for the specimens under alkaline, acidic, and neutral environments were 0.76%, 0.51% and 0.63% respectively; while the average volumetric contraction values were −1.37%, −0.79% and −1.14% (all values refer to the end of the 12 cycles, relatively to the initial volume). The images included in Figure 11 also show some superficial cracks in the specimens, as a consequence of the deterioration of the specimens at the end of the tests [40]. The specimens subjected to the alkaline and neutral environments showed a common behaviour at the beginning of their respective durability tests, as both showed a significant volume expansion during cycle # 2. Such volume increase at the initial stages of these tests was surely the cause of the superficial erosion of the specimens, as a result of internal stresses, which deteriorated the microstructure of the compacted soil. This is also behind the pronounced volumetric contraction observed in these two cases during cycles # 3 (alkaline medium) and # 9 (neutral medium).

The volumetric change evolution of the specimens subjected to the alkaline medium is apparently more irregular than the other two conditions, which might be related with the presence of soluble salts that formed during the test–the alkaline medium is prepared by dissolving sodium carbonate in water until a pH of 10 is reached. This procedure increases the probability of soluble expansive salts (carbonates) forming inside the stabilised soil voids, causing internal tensions and originating structural cracks [41].

Figure 12 shows the progressive loss of mass of each specimen during the 12 cycles. The accumulated mass loss, at the end of the 12 cycles, was 12.60%, 14.64 and 17.80%, for the specimens submerged in an alkaline, acidic and neutral environment, respectively. It can be seen that the specimen subjected to the acidic medium endured only 10 durability cycles, before its structure collapsed. This behaviour is a result of the acid attack, which progressively weakened the recently formed bond between the soil particles. On the other hand, the formation of soluble sulphates inside this specimen is also possible, as a consequence of the interaction between the acidic medium and the composition of the cementitious 50FA50WG paste. Indeed, the alkaline activator used is mainly composed of sodium and aluminium hydroxides, while the acidic medium was obtained using sulfuric acid. Most likely, a combination of the these two factors was responsible for the formation of both superficial and internal cracks in the stabilised soil mass, leading to the premature collapse of the specimen [42,43].

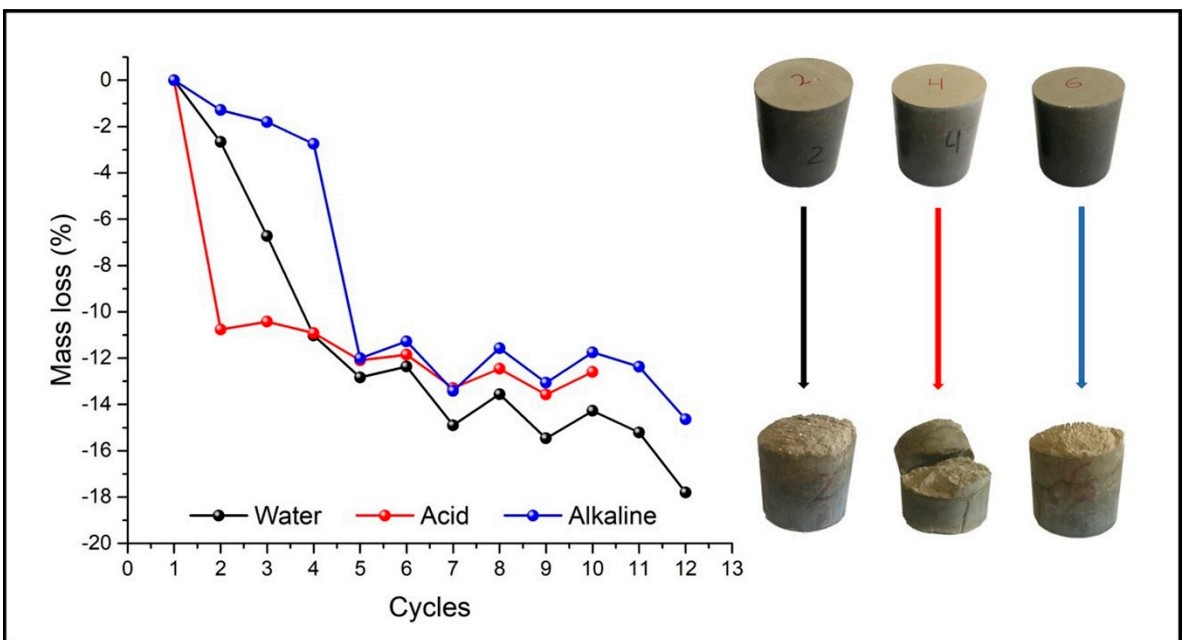

**Figure 12.** Mass loss throughout the wetting/drying cycles of 50/50 soil+binder combination, using an alkaline, an acidic and a neutral solution during the wetting stage.

### 3.2.4. Environmental Performance of the Selected Soil-Cement Combination

The soil stabilised with the 50FA50WG paste was submitted to a leaching test after 28 days curing. The elemental composition of the eluate obtained is presented in Figure 13, together with the limits for deposition on either inert waste and non-hazardous *waste* landfills, according with the EU Council Decision 2003/33/CE [27]. All values are below the *non-hazardous'* limits, and only the lead value is slightly above the corresponding *inert waste* limit. This is not surprising, given the relatively high lead content (5.07 wt%) of one of the precursors used, i.e., the glass waste, compared with the remaining heavy metal contents in all the wastes used.

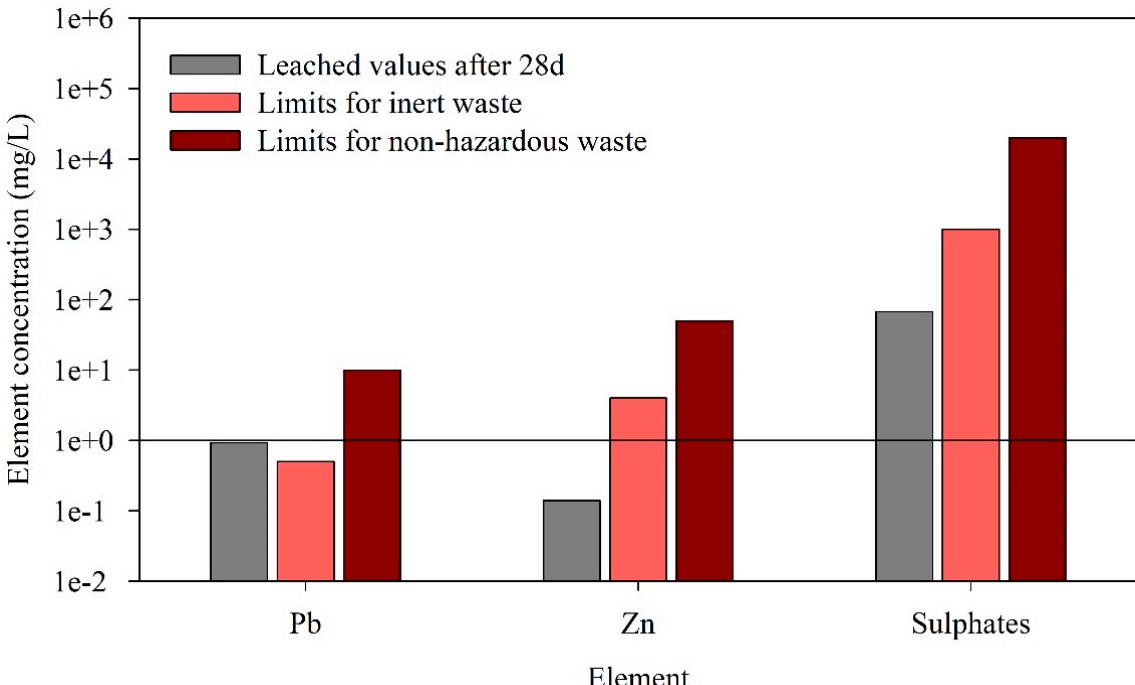

**Figure 13.** Element concentrations in the leaching solution retrieved from a specimen of soil stabilised with the 50FA50WG paste (using the EN 12457-4 test) and comparison with the acceptance limits defined in 2003/33/EC for inert and non-hazardous waste.

### 4. Conclusions

The results obtained in the present study show that it is possible to fabricate a binding material using solely industrial by-products–both as an activator and as a precursor, the two main components of alkaline activate cements. It was also shown that such binder can effectively be used to stabilise soft soils.

Several formulations were assessed, using different weight ratios between the two materials considered for the precursor–fly ash (FA) and glass waste (GW)–namely 100/00, 75/25, 50/50, 25/75 and 00/100; and different weight ratios between the activator–a recycled cleaning alkaline solution, and the solids (precursor), ranging from 0.4 to 0.85. The following conclusions were drawn:

- The compressive strength data revealed that an increase in the FA/GW ratio and a decrease in the *Activator/Precursor* ratio are beneficial.
- This trend was confirmed by the stiffness evolution during the first weeks, using seismic waves. There was a progressive increase in the P-wave velocity as curing progressed–an indication of the progressive increase of the elasticity modulus of the pastes–which was more intense for the pastes with higher fly ash content (100FA and 50FA/50WG).

- The reliability and precision of the correlation between the numerical and experimental data depends on the precursor composition. Therefore, the most effective soil-binder combinations were: 100FA (with S/P = 2.33 and A/S = 0.100) and 50FA50WG (with S/P = 2.33 and A/S = 0.115).
- The durability tests revealed mass loss values of 17.80%, 14.64% and 12.60%, under acid, neutral and alkaline media, respectively. The acid Environment was the most detrimental, suggesting even the possible structural collapse of the material.
- Leaching tests showed that the presence of Pb and Zn on the glass waste does not pose a problem, as the leached values of these elements are below the threshold values imposed by the mandatory standards and regulations.

**Author Contributions:** Conceptualization, N.C. and A.F.-J.; methodology, J.R.; validation, N.C. and A.F.-J.; investigation, J.R.; writing—original draft preparation, N.C. and J.R.; writing—review and editing, N.C. and A.F.-J.; project administration, N.C.; funding acquisition, N.C. and T.M. All authors have read and agreed to the published version of the manuscript.

**Funding:** This work was funded by the R&D Project JUSTREST- Development of Alkali Binders for Geotechnical Applications Made Exclusively from Industrial Waste, with reference PTDC/ECM-GEO/0637/2014, financed by the Foundation for Science and Technology—FCT/MCTES (PIDDAC).

**Institutional Review Board Statement:** Not applicable.

**Informed Consent Statement:** Not applicable.

**Data Availability Statement:** The study did not report any data.

**Acknowledgments:** The authors would like to acknowledge the contribution of the company Hydro Aluminium Extrusion Portugal HAEP SA, for the supply of the recycled cleaning solution; and the company POLO—Produtos Óticos SA, for the supply of the glass power.

**Conflicts of Interest:** The authors declare no conflict of interest. The funders had no role in the design of the study; in the collection, analyses, or interpretation of data; in the writing of the manuscript, or in the decision to publish the results.

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
