# Peer review of "Stabilisation of a Plastic Soil with Alkali Activated Cements Developed from Industrial Wastes"

_sustainability, doi:10.3390/su13084501_

Round 1

Reviewer 1 Report

There is a grammar mistake in the abstract (please avoid next time). Should that be "focuses" or "focused" instead of "focus" in Line 5 of abstract? Some language here reads a little bit strange. Please modify.

Why to measure fly ash particle size distribution? It is incredible to see the fly ash sieve analysis result in Figure 2. Did the authors fully dry and crush the fly ash before testing? This fly ash is more like a type of clay.

How to properly explain the decrease in P-wave velocity of the 100%WG group in Figure 8?

In Table 4, why the optimum moisture contents increased with decreasing the FA content and increasing the WG dosage?

Was the effect of soil suction considered in this study?

When conducting the wet-dry durability tests, how many replicated samples were prepared and tested for each set of mixture? Could the authors show the result variations? The reviewer is curious about the result representativeness. 

It is recommended to read carefully again the Introduction and Conclusions, to see if the key findings are associated tightly with the purpose and objectives of this study. If not, please modify. 

Reviewer 2 Report

The paper investigates to use alkali activated cements containing industrial wastes as a plastic soil stabilizer. There are following questions:

  1. Abstract need to be rewritten to report about the main and new findings obtained in this paper briefly.
  2. Many "Error! Reference source not found" appear in the manuscript.
  3. It needs to be clearly stated the contributions of the manuscript in the introduction section.
  4. The authors should provide the mechanism and theories for explanations.
  5. The author should compare the results with others’ research to confirm the contribution of this research.
  6. Conclusion is too long, should be described point by point and more precise and clearly.
